# Modulation Effect of Physical Activity on Sleep Quality and Mental Hyperactivity in Higher-Education Students

**DOI:** 10.3390/healthcare13091040

**Published:** 2025-05-01

**Authors:** Rubén Fernández-García, Cristina González-Forte, José Granero-Molina, Eduardo Melguizo-Ibáñez

**Affiliations:** 1Department of Nursing, Physiotherapy and Medicine, University of Almeria, 04120 Almeria, Spain; sirc98al@gmail.com (C.G.-F.); jgranero@ual.es (J.G.-M.); 2Facultad de Ciencias de la Salud, Universidad Autónoma de Chile, Santiago 7500000, Chile; 3Department of Specific Didactics, University of La Laguna, 38200 La Laguna, Spain; emelguiz@ull.edu.es

**Keywords:** physical activity, mental hyperactivity, sleep quality, higher education

## Abstract

**Objectives:** The present study seeks to analyze the relationships between the intensity of physical activity, mental hyperactivity and sleep quality. A comparative, descriptive and exploratory study was carried out. **Methods:** A sample of 1907 university students belonging to the degree of Physiotherapy and Physical Activity and Sport Sciences was used. The International Physical Activity and Mental Hyperactivity Questionnaires were used. The scale used was the Pittsburgh sleep quality index. The proposed model analyzes the relationships of physical activity with mental hyperactivity and various sleep-related factors. **Results:** The following fit indices were evaluated: Chi-Square = 80.242; Degrees of Freedom = 3; Incremental Fit Index = 0.951, Comparative Fit Index = 0.977; Normed Fit Index = 0.946; Root Mean Square Error of Approximation = 0.071. The values obtained show the good fit of the theoretical model. Statistically significant differences are observed (*p* < 0.05) in the causal relationship of mental hyperactivity with the personal assessment of sleep as a function of the intensity of physical activity. A greater effect of light (β = 0.671) compared to moderate- (β = 0.428) or vigorous-intensity (β = 0.343) physical activity in personal sleep assessment is evident. Statistically significant differences were also observed in the causal relationship of mental hyperactivity with the time to fall asleep (*p* < 0.05). Light physical activity (β = 0.479) has a greater causal relationship with time to fall asleep than moderate- (β = 0.302) or vigorous-intensity (β = 0.413) physical activity. **Conclusions:** Based on the results obtained, it is concluded that the intensity with which physical activity is performed has a modulating effect on sleep quality and mental hyperactivity.

## 1. Introduction

There is currently growing concern regarding the number of people with a tendency to be sedentary [1,2]. There are many detrimental effects associated with physical inactivity [3]. These include the elevated risk of cardiovascular problems, diabetes and cancer, among others [3]. To mitigate the risks of a sedentary lifestyle, many governmental, health and sports organizations have updated certain guidelines, encouraging the practice of regular physical activity. The World Health Organization [4] specifically advises that adults engage in 150 to 300 min of moderate-intensity physical activity (PA) or 75 to 150 min of vigorous-intensity PA each week. For children and adolescents, at least 60 min of moderate- or vigorous-intensity aerobic activity daily is recommended [4]. There are different factors to evaluate the degree of intensity at which a physical activity is performed. One of the most common and widely used is heart rate. An exercise is considered to be of mild intensity when it is performed at up to 65% of the maximum HR [4]. An exercise is considered to be of moderate intensity when conducted at up to 70% of the maximum HR [4]. An exercise is considered to be of vigorous intensity when carried out at up to 85% of maximum HR [4]. Another factor widely used to calculate the intensity of a physical activity is the METs. These are defined as a unit of measurement that quantifies energy expenditure and oxygen consumption during physical activity [4]. A physical activity is light when it has an expenditure between 1.5 and 3 METs [4]. A physical activity has a moderate intensity when it is performed between 3 and <6 times the resting intensity (METs) [4]. Finally, an activity is considered vigorous when it is performed at >3 METs [4]. Research has demonstrated that regular physical activity, particularly when designed by professionals like physiotherapists or certified personal trainers, is an effective preventive measure against chronic diseases and mental health issues [5,6,7,8,9,10].

Focusing on the adolescent population, there are numerous biological, cognitive and psychological changes that they undergo [11]. Studies on physical activity in adolescents attempt to elucidate the determinants of sports practice [12]. Although there is no consensus on the PA patterns of adolescents, studies have focused on a number of indicators such as type, frequency, duration and intensity [13]. Numerous studies indicate that it is not possible to obtain conclusive conclusions when only a single indicator is analyzed [14]. It is necessary to study several at the same time since the practice of physical activity is affected not only by biological variables but also by social, cultural and environmental variables [14]

One of the factors that usually worsen our health both in the short and long term is chronic stress [15,16,17]. University adolescents are a type of population that is undoubtedly subjected to a multitude of stressors [18]. The practice of physical activity is a protective variable against the harmful effects of stress enhancers that the consumer society promotes [19,20,21].

The brain is composed of a multitude of complex networks, consisting of functionally interconnected regions that communicate dynamically with each other [22]. These neurocognitive networks contain groups of neurons from interconnected cortical areas, with the purpose of mediating specialized cognitive performances and anatomically different structures [22]. One such region is the default neural network, which shows greater activity at rest than during task performance [23,24]. A relationship between the activation levels of this network and different pathologies such as Alzheimer’s disease, schizophrenia, autism, depression and attention deficit disorder [25] and with emotional disorders such as traumatic stress disorder [26], trait anxiety [27] and depression [28] has been demonstrated. The default neural network, characterized by low neuronal density, also plays an important role in personal abstraction [28]. This network significantly influences the construction of such relevant emotions as happiness and sadness [29]. Increased activity on the part of the default neural network generates a state of mental hyperactivity, facilitating states of rumination or an increase in the number of mental images connected to an excess of thoughts [30]. Mental hyperactivity is defined as a pattern of excessive and sustained cognitive activation, characterized by a high frequency of spontaneous thoughts, ruminations or undirected mental associations, which can interfere with attentional processes, emotional regulation and psychobiological rest [22]. This state has been linked to the hyperconnectivity of brain networks such as the Default Mode Network (DMN), particularly in conditions of anxiety, insomnia and affective disorders [22]. It is necessary today to develop assessment methods, as well as effective techniques and procedures, that help to manage mental hyperactivity [22]. In this regard, the practice of physical activity and exercise represents an excellent option [31].

The term sleep quality can be defined as the ability to fall asleep easily and also experience deep sleep [32]. Currently, more than 50% of people worldwide are affected by sleep disorders [31,32]. A total of 33% of adults have difficulty falling asleep or staying asleep all night or wake up tired with a feeling of having had their sleep disturbed [32]. Good-quality sleep is vital for the general population and specifically in the university context. In the latter case, if sleep is not restful, in the long term, some of the consequences can be poor academic performance [32], anxiety problems, depression and even suicidal thoughts [33]. The practice of physical activity can be a good strategy to promote sleep quality. The release of interleukin-1 and -6 during moderate-intensity exercise can facilitate sleep [34]. The release of brain-derived neurotrophic factor during physical activity improves memory processing during sleep [35,36]. Finally, growth hormone secretion due to physical exercise is associated with slow-wave sleep [37].

Given the above, the following research hypotheses are suggested:

**H1.** 
*The level of intensity at which physical activity is carried out influences the connection between mental hyperactivity and individuals’ evaluation of their sleep.*


**H2.** 
*Engaging in light-intensity physical activity has a positive association with mental hyperactivity and the time it takes to fall asleep.*


**H3.** 
*The intensity of physical activity is linked to increased medication use and higher levels of sleep disturbance.*


The main objective of the study is as follows:

**O1.** 
*To analyze the relationship between physical activity, mental hyperactivity and the different amounts and qualities of sleep.*


## 2. Materials and Methods

### 2.1. Design and Participants

The steps outlined in the STROBE statement [38] were used as a reference. The research was exploratory, descriptive, cross-sectional and comparative. The initial sample included 1950 participants, of which 43 were excluded for not responding appropriately to the survey items. The final sample comprised 1907 students (mean age = 30.49; standard deviation = 2.35) from various Andalusian universities. According to the distribution by sex, 787 belong to the male sex (41.26%) and 1120 to the female sex (58.74%). Non-probability and convenience sampling methods were employed. The sampling error for this study was 0.022, with a 95% confidence interval. In this case, the students belong to the social sciences area of study. The inclusion criteria were being a university student and being over 18 years of age, with no exclusion criteria impacting the analysis.

### 2.2. Instruments and Variables

The Sociodemographic Questionnaire included questions related to the age and sex of the participants (male/female).

The International Physical Activity Questionnaire tool gathers data on the time and frequency spent on activities of different intensities [39]. It includes seven questions with reliable measurement properties for assessing physical activity levels across different contexts. The questionnaire evaluates three aspects of physical activity: intensity (light, moderate or vigorous), frequency (days per week) and duration (times per day) [40]. The instrument has demonstrated reliability in previous research involving university students [41]. Adequate reliability values were obtained (α = 0.77; ω = 0.78).

The Mental Hyperactivity Questionnaire [22] evaluates indirectly, through the completion of only 10 items, the activation of the default neural network during the last three months. This questionnaire has only been validated in Spanish [22]. It has also been validated in a university population [22]. Adequate reliability values were obtained for this instrument (α = 0.89; ω = 0.89).

A version of the Pittsburgh sleep quality [42] index adapted to Spanish [43] was used. It evaluates 19 items, including subjective sleep quality, daytime dysfunction, duration, efficiency, latency and sleep disturbance. It also evaluates medication consumption. The values of this scale were α = 0.78 and ω = 0.80.

### 2.3. Procedure

Considering the instruments needed to carry out the research, Google Forms was used to create a questionnaire with all the items of interest. The university students who met the inclusion criteria filled in the questionnaire through a link they received in an e-mail. To verify the accuracy of the questionnaire responses, three items were repeated. If the answers to these items did not match, the participant’s data were excluded. Data collection occurred between April and December 2024. During data collection, the students were subjected to a period of evaluative testing (June 2024). Data collection also coincided with the end of a term (February–June of the 2023–2024 academic year and the beginning of the September–January term of the 2024–2025 academic year). All participants voluntarily took part after providing informed consent. The study adhered to the ethical guidelines set forth in the Declaration of Helsinki and was consistently overseen by the University of Granada’s ethics committee.

### 2.4. Data Analysis

The IBM SPSS v29.0.2 statistical package was used to perform the statistical analysis of the results. Normality was assessed using the skewness and kurtosis values for each item. These values were expected to meet the standard normality criteria: skewness should fall between −1.5 and 1.5 [44,45], and kurtosis should range from −3 to 3 [44,45]. Additionally, the reliability of the instruments was evaluated using Cronbach’s Alpha and McDonald’s Omega tests, with the reliability index set at 95%.

The IBM AMOS v23 statistical package was used to develop the multigroup equation model. To assess the model fit, the values of the Incremental Fit Index (IFI), Comparative Fit Index (CFI) and Normalized Fit Index (NFI) were examined. For a good fit, these indices should be greater than 0.90 [46]. Additionally, the Root Mean Square Error of Approximation (RMSEA) was considered, with values below 0.08 indicating a good fit [47].

Figure 1 presents the theoretical model showing the direction of the causal relationships of the variables. The theoretical model is made up of 4 endogenous variables and 1 exogenous variable. Unidirectional causal relationships are observed. These occur when one-way arrows appear, with the independent variable being the origin of the arrow and the tip being the dependent variable.

The mental hyperactivity variable acts as an exogenous variable. It exerts a unidirectional effect on the personal assessment of sleep, time of sleep conciliation, sleep disturbance and the use of sleep medication. Likewise, a unidirectional effect of mental hyperactivity on bodily pain is observed. All these variables are continuous and were calculated through the mean value of the items of each dimension of the instrument.

After proposing and developing the theoretical model of the research, the next step was to evaluate its fit. The fit indices evaluated were as follows: X^2^ = 80.242; gl = 3; IFI = 0.951, CFI = 0.977; NFI = 0.946; RMSEA = 0.071. The values obtained show the good fit of the theoretical model [46,47].

## 3. Results

Figure 2 presents the theoretical model with the regression weights for the total sample. A positive causal relationship of mental hyperactivity with personal perception of sleep (β = 0.67), sleep conciliation time (β = 0.48), use of medication (β = 0.10) and sleep disturbance (β = 0.350) is observed. A positive effect of personal assessment of sleep-on-sleep disturbance is also observed (β = 0.26). A positive effect of sleep reconciliation time on the use of medication was obtained (β = 0.11). A positive causal relationship of personal perception of sleep on medication use is observed (β = 0.045). Finally, a positive effect of the time to fall asleep on sleep disturbance was obtained (β = 0.054).

Table 1 displays the mean values, standard deviations, skewness and kurtosis for each variable. It also includes the correlation matrix for the variables included in the structural equation model. The skewness and kurtosis values are within acceptable ranges for all variables, meeting standard normality criteria. Skewness values range from −1.5 to 1.5, and kurtosis values are between −3 and 3, indicating that the sample data follow a normal distribution [44,45].

Table 1 also presents the correlational analysis of the variables. An *r* value between 0 and 0.1 indicates no correlation [48]. A value between 0.1 and 0.29 shows a low correlation [48]. A value between 0.3 and 0.49 shows a medium association [48]. A value between 0.5 and 0.69 denotes a high correlation [48]. Finally, a value between 0.70 and 1 indicates a very strong correlation [48]. In this case, a medium and significant correlation between mental hyperactivity and personal assessment of sleep (r = 0.449; *p* < 0.01), use of medication (r = 0.319; *p* < 0.01), sleep conciliation time (r = 0.376; *p* < 0.01) and sleep disturbance (r = 0.494; *p* < 0.01) was observed. Regarding personal assessment of sleep, there was a medium and significant correlation with the use of medication (r = 0.315; *p* < 0.01) and sleep conciliation time (r = 0.464; *p* < 0.01). A strong correlation was observed between the variables personal assessment of sleep and sleep disturbance (r = 0.590; *p* < 0.01). Continuing with the variable use of medications, a medium correlation is shown with sleep conciliation time (r = 0.256; *p* < 0.01). On the other hand, a strong correlation was observed between the use of medications and sleep disturbance (r = 0.517; *p* < 0.01). Finally, a medium correlation was observed between sleep conclusion time and sleep disturbance (r = 0.382; *p* < 0.01).

Table 2 presents the standardized regression weights for the variables analyzed based on physical activity intensity. Statistically significant differences (*p* < 0.05) are found in the causal relationship between mental hyperactivity and personal sleep assessment, depending on the intensity of physical activity. A stronger effect is seen with light physical activity (β = 0.671) compared to moderate (β = 0.428) or vigorous (β = 0.343) activity. Statistically significant differences are also observed in the relationship between mental hyperactivity and sleep onset time (p < 0.05). Participants engaging in light physical activity (β = 0.479) show a stronger causal link than those performing moderate- (β = 0.302) or vigorous-intensity (β = 0.413) activity. Lastly, significant differences are found in the impact of using sleep medications on sleep disturbance (*p* < 0.05), with a greater effect seen in participants engaging in vigorous physical activity (β = 0.393) compared to those partaking in moderate- (β = 0.298) or light-intensity (β = 0.252) exercise.

## 4. Discussion

The present investigation evaluated the importance of physical intensity with respect to the cognitive variable mental hyperactivity (MH) and different factors related to sleep. Sleep helps in the consolidation of memory and learning.

The results indicated a causal relationship of mental hyperactivity with the personal assessment of sleep as a function of the intensity of physical activity. It was found that people who undertake light physical activity show a greater tendency toward mental hyperactivity and report improved sleep. This finding is first explained by the fact that evaluations of oneself or others, related to sleep in this specific case, depend on the functioning of certain areas of the brain. One of these areas is the default neural network, which shows greater activity at rest than during task performance [23] and is directly involved in MH processes. Tasks involving greater attention further deactivate the default neural network and reduce mental hyperactivity [22]. If the activity affects the functioning of the default neural network, there is a greater chance that more thoughts will occur at light intensity because the brain areas involved in attention are less activated [49]. On the other hand, the default neural network is activated mainly before the person encounters stressors considered a problem [50]. If someone has problems with sleep and this situation generates mental stress, it is very likely that they will have more thoughts related, in this case, to the personal assessment of sleep.

Statistically significant differences were observed in the causal relationship of mental hyperactivity with the time to fall asleep. Participants that engaged in light physical activity had a stronger causal relationship than those that engaged in physical activity at moderate or vigorous intensity. The interpretation of the results can be approached from the perspective of the cortisol hypothesis [51]. Cortisol is an adrenal hormone that is released in response to stress. It presents high levels in the morning and a progressive decline throughout the day [51]. The human body continuously responds to internal and external stressors. Depending on the degree of threat, an organism processes one response or another. The response generates the activation of a whole series of hormonal and physiological responses. The amygdala and hypothalamus are two main structures in the stress response [52]. As the body continues to perceive stimuli as a threat, the hypothalamus activates the hypothalamus–pituitary–adrenal axis to maintain a state of maximum alertness and to facilitate through cortisol the energy necessary for proper organ function [53]. Cortisol is maximal in the morning and progressively decreases. In the general population, cortisol levels are becoming increasingly high in the evening. Many of the stressors that affect us are mental and prevent the activation of the parasympathetic system or the system of physical and mental relaxation. This situation has a negative influence when it comes to falling asleep since stress, cortisol and energy levels are very high and prevent us from falling asleep. In this sense, high levels of stress and energy facilitate mental hyperactivation and an increase in the number of discriminated thoughts, which inhibit sleep to a great extent [54,55]. Considering this explanation, the results obtained indicate that light physical activity is not sufficiently effective in facilitating sleep. Moderate or vigorous physical activity is necessary to regulate cortisol and high energy levels due to mental stress [56].

Finally, statistically significant differences were found in the effect of the use of medication to fall asleep on sleep disturbance. A greater effect was observed for participants who performed vigorous physical activity compared to those who performed moderate or light physical activity. First, a possible explanation for the results obtained is the time at which the physical activity is performed. If the physical activity is carried out shortly before bedtime and is also of vigorous intensity, numerous studies have shown that it can affect the different characteristics of sleep [57]. This explanation is not sufficient since not all people who partake in vigorous sport take drugs to improve sleep. Secondly, it is likely that people who consume drugs to be able to sleep are not only affected by partaking in vigorous sport shortly before bedtime but also almost certainly by other modulating variables. In this sense, the increased consumption of drugs may be due to two variables that affect sleep more than physical activity itself: stress and anxiety [57]. It is also possible that the most stressed people are precisely those who practice sport at a higher intensity to control the effects of stress and mental hyperactivity. In this sense, the practice of effective techniques to help manage emotions and relax the mind is advised [58,59].

## 5. Limitations and Perspective Futures

Although the number of participants included was representative, it is always valuable to include as many as possible. It would be worthwhile to increase the sample size in future studies. We focused on two very specific populations of university students. It would be interesting in future research to obtain students belonging to other branches of knowledge and to carry out comparative studies according to the degree that the participants are studying. Similarly, it would also be an excellent contribution to obtain questionnaire and scale results from university professors from different degrees, with the aim of comparing students and teachers. This study may be affected by bias due to the lack of control of an extraneous variable related to the sending of the test battery via an e-mail link. Although the instructions were very clear, and the questionnaires were very easy to fill out, this does not preclude the possibility that some people made mistakes in answering the items. The non-probabilistic and convenience registration when collecting the sample should also be noted as a limitation. This study clarified the direct relationships between the intensity of physical activity, mental hyperactivity and sleep quality. In this sense, the possibility of including new variables in addition to those already used, such as physical pain, life satisfaction, anxiety levels and depression, is being considered.

## 6. Conclusions

Based on these results, it is concluded that the intensity at which physical activity is carried out conditions the causal relationships of mental hyperactivity with variables related to sleep quality. Based on the conclusions obtained, the need to condition physical activity in university students according to their exercise intensity and level of mental hyperactivity is highlighted.

## Figures and Tables

**Figure 1 healthcare-13-01040-f001:**
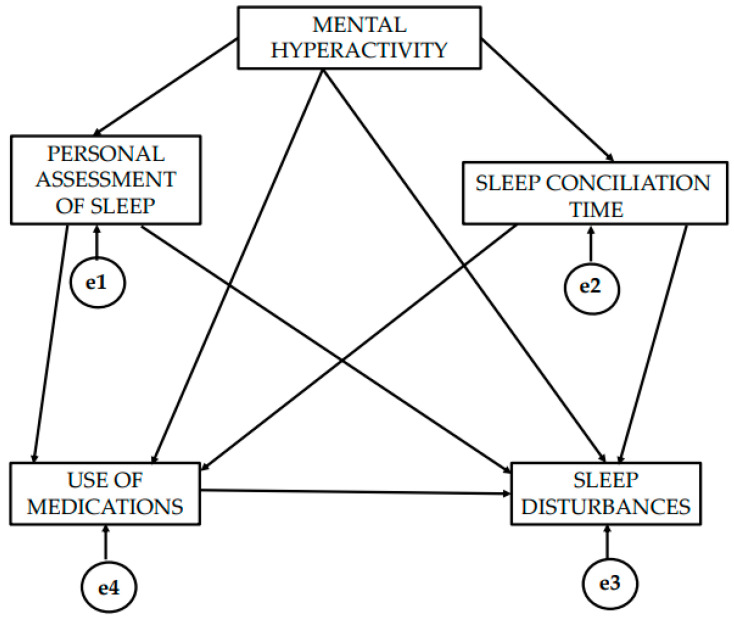
Theoretical representation of the structural equation model.

**Figure 2 healthcare-13-01040-f002:**
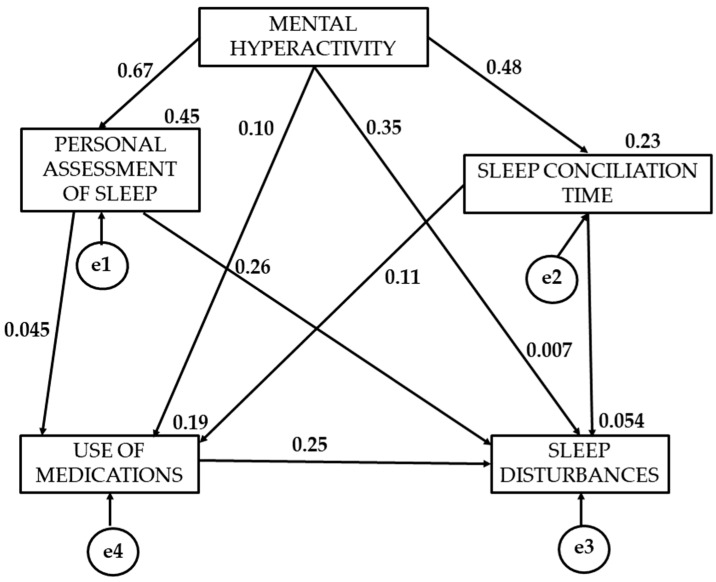
Theoretical model with standardized regression weights.

**Table 1 healthcare-13-01040-t001:** Descriptive statistics, reliability and correlation analyses of the study variables.

	M	SD	SWEK	KUR	2	3	4	5
**1. MH**	1.21	0.69	0.390	−0.507	0.449 **	0.319 **	0.376 **	0.494 **
**2. PAS**	1.14	0.72	0.381	0.146		0.315 **	0.464 **	0.590 **
**3. UM**	0.30	0.78	0.652	0.810			0.256 **	0.517 **
**4. SCT**	1.88	0.88	0.684	0.424				0.382 **
**5. SD**	0.74	0.42	0.850	0.906				

**Note:** ** *p* < 0.01; mental hyperactivity: MH; personal assessment of sleep: PAS; use of medications: UM; sleep conciliation time: SCT; sleep disturbance: SD.

**Table 2 healthcare-13-01040-t002:** Standardized causal relationship of the variables.

Direction of Causal Relationships		Regression Weights				Standardized Regression Weights
		Estimate	Estimation Error	Critical Radio	*p*	β
MILD PA	MH → PAS	0.735	0.66	11.143	0.049	0.671
MODERATE PA	MH → PAS	0.426	0.057	7.525		0.428
VIGOROUS PA	MH → PAS	0.364	0.075	4.828		0.343
MILD PA	MH → SCT	0.602	0.090	6.714	0.024	0.479
MODERATE PA	MH → SCT	0.391	0.078	5.036		0.302
VIGOROUS PA	MH → SCT	0.531	0.088	6.006		0.413
MILD PA	MH → UM	0.392	0.101	3.892	0.073	−0.096
MODERATE PA	MH → UM	0.158	0.064	2.455		0.169
VIGOROUS PA	MH → UM	−0.123	0.136	−0.903		0.309
MILD PA	PAS → UM	0.168	0.087	1.929	0.054	0.449
MODERATE PA	PAS → UM	0.148	0.062	2.381		0.158
VIGOROUS PA	PAS → UM	0.523	0.115	4.540		0.141
MILD PA	UM →SCT	0.075	0.074	1.014	0.310	0.115
MODERATE PA	UM →SCT	0.063	0.045	1.397		0.088
VIGOROUS PA	UM →SCT	0.117	0.085	1.377		0.077
MILD PA	MH → SD	0.035	0.028	1.250	0.211	0.352
MODERATE PA	MH → SD	0.096	0.031	3.037		0.162
VIGOROUS PA	MH → SD	0.225	0.052	4.355		0.193
MILD PA	TCS → SD	0.035	0.028	1.25	0.076	0.075
MODERATE PA	TCS → SD	0.004	0.022	0.168		0.008
VIGOROUS PA	TCS → SD	0.038	0.032	1.176		0.072
MILD PA	VPS → SD	0.197	0.033	5.971	0.258	0.264
MODERATE PA	PAS → SD	0.266	0.030	8.747		0.448
VIGOROUS PA	VAS → SD	0.154	0.047	3.315		0.338
MILD PA	UM → SD	0.192	0.028	6.789	0.006	0.252
MODERATE PA	UM → SD	0.187	0.030	6.175		0.298
VIGOROUS PA	UM → SD	0.126	0.031	4.098		0.393

**Note:** mental hyperactivity: MH; personal assessment of sleep: PAS; use of medications: UM; sleep conciliation time: SCT; sleep disturbance: SD.

## Data Availability

The data used to support the findings of the current study are available from the corresponding author upon request.

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
