# Peer review of "Modulation Effect of Physical Activity on Sleep Quality and Mental Hyperactivity in Higher-Education Students"

_healthcare, 2025, doi:10.3390/healthcare13091040_

Round 1

Reviewer 1 Report

Comments and Suggestions for Authors

The main objective of the study was to analyze the relationship between physical activity, mental hyperactivity and different amounts and quality of sleep. The following is suggested in the manuscript to clarify certain aspects.

Abstract

21-22 Clarify what (which parameter) is more influenced by light physical activity compared to moderate and intensive physical activity.

22 - the letter S is missing in the word  „Statistically”

24-26 – It is important to clarify what the cause-effect relationship refers to. I propose to write explicitly that participants undertaking light physical activity (fell asleep faster/slower) than those who did moderate or high intensity physical activity.

26-28 In order to maintain linguistic consistency, I would use temporal formulation, i.e.: Based on the results obtained, it is concluded that the intensity with which physical activity is performed has a modulating effect on sleep quality and mental hyperactivity.

Introduction

Lack of clear definition of ‘metal hyperactivity’, what researchers mean by using this phrase.

Better structuring of the content in the presentation of the argumentation is required, it is worth describing the problems in more detail, i.e. mental hyperactivity (definition, what influences it, what are the symptoms, which population is affected both locally and worldwide) and sleep quality (give determinants of sleep quality and epidemiology).  Describe arguments and examples from the literature on how physical activity affects the above problems. Avoid sentences that are not directly relevant to the manuscript, e.g:

44-47 excerpt not related to the topic of the paper.

Methodology

107 - ultimately from how many academic units the respondents came from

110 - was the field of study relevant for inclusion criteria in the study?

How were reliability values measured?

Did the questionnaire include a socio-demographic part? - if yes please describe this part of the questionnaire as well.

To how many email addresses was the questionnaire sent?

Where did the database of e-mail addresses come from?

What was the time required to complete a single questionnaire?

How is the stored data secured?

Discussion

210-221 – this is a fragment more suitable for an introduction. The discussion should contain references and connections between the obtained results and the results of other researchers in a specific scientific area.

Cocnclusions – There are no conclusions in this section, only results are presented.

Author Response

Comment 1

21-22 Clarify what (which parameter) is more influenced by light physical activity compared to moderate and intensive physical activity.

Response 1

Thank you very much for your comment. The authors have rephrased the sentence

Comment 2

22 - the letter S is missing in the word  „Statistically”

Response 2

Thank you for your comment. The authors have added the S

Comment 3

24-26 – It is important to clarify what the cause-effect relationship refers to. I propose to write explicitly that participants undertaking light physical activity (fell asleep faster/slower) than those who did moderate or high intensity physical activity.

Response 3

Thank you very much for your comment. The authors have added the missing information.

Comment 4

26-28 In order to maintain linguistic consistency, I would use temporal formulation, i.e.: Based on the results obtained, it is concluded that the intensity with which physical activity is performed has a modulating effect on sleep quality and mental hyperactivity.

Response 4

Thank you for your comment. The authors have replaced the initial sentence with the one you propose.

Comment 5

Lack of clear definition of ‘metal hyperactivity’, what researchers mean by using this phrase.

Response 5

Thank you very much for your comment. The concept of mental hyperactivity has been defined.

Comment 6

Better structuring of the content in the presentation of the argumentation is required, it is worth describing the problems in more detail, i.e. mental hyperactivity (definition, what influences it, what are the symptoms, which population is affected both locally and worldwide) and sleep quality (give determinants of sleep quality and epidemiology).  Describe arguments and examples from the literature on how physical activity affects the above problems. Avoid sentences that are not directly relevant to the manuscript, e.g:

Response 6

Thank you very much for your comment. The introduction has been reworded

Comment 7

107 - ultimately from how many academic units the respondents came from

Response 7

Thank you very much for your comment. The students only belong to the social sciences area of study.

Comment 8

110 - was the field of study relevant for inclusion criteria in the study?

Response 8

Thank you very much for your comment. This question matches another reviewer's question. We, the authors, have proceeded to add the area of study to which the students belong. For convenience, the authors established that the participants belonged to the area of social sciences.

Comment 9

How were reliability values measured?

Response 9

Thank you very much for your comment. Reliability has been studied through Cronbach's Alpha and McDonald's Omega.

Comment 10

Did the questionnaire include a socio-demographic part? - if yes please describe this part of the questionnaire as well.

Response 10

Thank you very much for your comment. We have added the requested information.

Comment 11

To how many email addresses was the questionnaire sent?

Response 11

Thank you very much for your comment. To send the link to the questionnaire, the mailing lists were used. In this case and after consulting the list, the initial message together with the questionnaire was sent to 4329 students in the social sciences area.

Comment 12

Where did the database of e-mail addresses come from? What was the time required to complete a single questionnaire? How is the stored data secured?

Response 12

The diffusion list belonged to the University of Almeria. The research team contacted the University's dissemination team. Once the appropriate permission and ethics committee was obtained, the message was sent to the participants. The questionnaire took between 8 and 10 minutes to complete. Also, to ensure data protection and to maintain the anonymity of the participants, participants were asked to identify themselves with their date of birth and the initial of their first surname (30/06/1998M). Only the research team has access to the database.

Comment 13

210-221 – this is a fragment more suitable for an introduction. The discussion should contain references and connections between the obtained results and the results of other researchers in a specific scientific area.

Response 13

Thank you very much for your comment. The authors have removed the requested information.

Comment 14

Conclusions – There are no conclusions in this section, only results are presented.

Response 14

Thank you very much for your comment. The conclusions have been reformulated

Reviewer 2 Report

Comments and Suggestions for Authors

Dear authors,

Thank you for submitting your article, which I found both very interesting and well-written. The effort made to ensure scientific rigor is obvious, notably through the explicit reference to the stages of the STROBE declaration, which testifies to an appreciable concern for methodological transparency.

The article stands out for its robust statistical approach, with original modeling using IBM AMOS v23 software, in addition to SPSS v29.0.2. This methodological choice, relatively uncommon in sport and health science publications, deserves to be highlighted. In addition, the article is well documented, with 69 relevant references.

However, despite these qualities, certain limitations caught my attention and deserve to be corrected in order to strengthen the scope and interpretability of your results:

Insufficient sample characterization:
It is regrettable that the article does not indicate the average age, standard deviation or distribution of students by gender, level of study or discipline. These variables are essential to contextualize the results, particularly in the field of academic stress. The stress of a first-year medical student is not comparable to that of a Master's student in philosophy or architecture, disciplines which also differ in the way they are assessed (exams, competitions, continuous assessment, etc.).
Time of survey in the academic year :
No information is given on the data collection period. However, stress levels and sleep quality vary greatly depending on whether it's the beginning of the semester or the exam period.

Typology of physical activities:
The nature of physical activities is not specified. However, the expected effects on stress and sleep quality can vary significantly between, for example, weight training, nature running, swimming, yoga or tai chi. A more detailed analysis of the types of activities practised would enrich your approach, notably via multivariate or multi-group analyses, as part of your structural equation modelling.

Summary to be clarified:
The summary contains several technical abbreviations (X², gl, IFI, CFI, RMSEA) that are not explained. It would be useful for non-specialist readers if these indices were named at least once. A summary should be comprehensible independently of the body of the text.

Clarification of physical activity intensities:
In lines 37-40, you mention light, moderate or vigorous physical activity. It would be relevant to briefly characterize them with usual physiological indicators such as heart rate, METs, Borg scale or speech test, to facilitate the reader's understanding.

In short, your study is rich and well-constructed, and makes an original contribution to understanding the relationships between physical activity, stress and sleep in students. I would simply like to see the above-mentioned elements integrated or clarified, in order to strengthen the scope of your work.

Good luck with the rest of the revision process.

Yours faithfully

Author Response

Comment 1

Dear authors,

Thank you for submitting your article, which I found both very interesting and well-written. The effort made to ensure scientific rigor is obvious, notably through the explicit reference to the stages of the STROBE declaration, which testifies to an appreciable concern for methodological transparency.

The article stands out for its robust statistical approach, with original modeling using IBM AMOS v23 software, in addition to SPSS v29.0.2. This methodological choice, relatively uncommon in sport and health science publications, deserves to be highlighted. In addition, the article is well documented, with 69 relevant references.

However, despite these qualities, certain limitations caught my attention and deserve to be corrected in order to strengthen the scope and interpretability of your results:

Response 1

Thank you very much for your comment. We will make the necessary changes to improve the quality of the research.

Comment 2

It is regrettable that the article does not indicate the average age, standard deviation or distribution of students by gender, level of study or discipline. These variables are essential to contextualize the results, particularly in the field of academic stress. The stress of a first-year medical student is not comparable to that of a Master's student in philosophy or architecture, disciplines which also differ in the way they are assessed (exams, competitions, continuous assessment, etc.).

Response 2

Thank you very much for your comment. The authors have added the area of study to which the students belong. 100% of the students belong to the area of social sciences. This is because the research team had greater access to this sample.

Comment 3

No information is given on the data collection period. However, stress levels and sleep quality vary greatly depending on whether it's the beginning of the semester or the exam period

Response 3

Thank you very much for your suggestion for improvement. The authors have proceeded to add the requested information.

Comment 4

The nature of physical activities is not specified. However, the expected effects on stress and sleep quality can vary significantly between, for example, weight training, nature running, swimming, yoga or tai chi. A more detailed analysis of the types of activities practised would enrich your approach, notably via multivariate or multi-group analyses, as part of your structural equation modelling.

Response 4

Thank you very much for your comment. The authors fully agree with you. The instrument used to obtain the intensity of physical activity was the International Physical Activity Questionnaire. This questionnaire allows us to calculate the number of METs through the following questions: During the last 7 days, on how many days did you perform vigorous physical activities such as heavy lifting, digging, aerobics, or high-speed cycling; How much time did you normally spend performing vigorous physical activities on one of those days; During the last 7 days, on how many days did you perform moderate physical activities such as carrying light loads, cycling at a regular pace, or playing doubles tennis; During the last 7 days, on how many days did you perform moderate physical activities such as carrying light loads, cycling at a regular pace, or playing doubles tennis? During the last 7 days, on how many days did you walk for at least 10 minutes at a time; How much time did you usually spend walking on one of those days?

As a future perspective it would be advisable to ask about the nature of the physical activity the subjects engage in

Comment 5

The summary contains several technical abbreviations (X², gl, IFI, CFI, RMSEA) that are not explained. It would be useful for non-specialist readers if these indices were named at least once. A summary should be comprehensible independently of the body of the text.

Response 5

Thank you very much for your comment. The authors have removed the abbreviations and have added the full name.

Comment 6

In lines 37-40, you mention light, moderate or vigorous physical activity. It would be relevant to briefly characterize them with usual physiological indicators such as heart rate, METs, Borg scale or speech test, to facilitate the reader's understanding.

Response 6

Thank you very much for your comment. The authors have added the requested information.

Reviewer 3 Report

Comments and Suggestions for Authors

Introduction

Adequate and well-founded. Presents a broad overview of the effects of physical activity and sleep problems, relating it to mental hyperactivity. The references used are recent and relevant, and the text contextualizes the problem well.

Materials and methods

The descriptive, cross-sectional design is well defined, with a large sample. The instruments are adequately described, including reliability and validity. However: It is suggested to expand the justification for the use of the mental hyperactivity questionnaire, since it is a new tool developed by the authors and validated only in Spanish. The procedure section is clear, although the use of non-probability convenience sampling should be highlighted as a limitation.

Data analysis:

The statistical analysis is robust. Normality check, correlational analysis and structural equation models with adequate adjustment indexes are performed.

Results:

Results are presented in tables with correlation values, normality and standardized weights. Good interpretation of the effects according to the intensity of physical activity.

Discussion:

The findings are adequately related to previous studies, although there are speculative interpretations without direct measurement. It is recommended: to explicitly state that these are untested hypotheses and to reduce excessively narrative or theoretical paragraphs without direct empirical basis in the study.

Limitations, conclusions and bibliographical references:

Acknowledges limitations such as the type of sample and the online format of the questionnaire. It would be positive to highlight more the impossibility of establishing causality and to propose longitudinal or experimental studies as an alternative.

The conclusions are in line with the objectives and results. However, the statement on medication consumption by physically active people should be more cautious or nuanced, as there is no measurement of the time of exercise or associated clinical variables.

Relevant and updated references, recent and high impact citations.

Author Response

Comment 1

Introduction: Adequate and well-founded. Presents a broad overview of the effects of physical activity and sleep problems, relating it to mental hyperactivity. The references used are recent and relevant, and the text contextualizes the problem well.

Response 1

Thank you very much for your comment

Comment 2

Materials and methods: The descriptive, cross-sectional design is well defined, with a large sample. The instruments are adequately described, including reliability and validity. However: It is suggested to expand the justification for the use of the mental hyperactivity questionnaire, since it is a new tool developed by the authors and validated only in Spanish. The procedure section is clear, although the use of non-probability convenience sampling should be highlighted as a limitation.

Response 2

Thank you very much for your comment. The non-probabilistic and convenience registration has been pointed out as a limitation.

Comment 3

Data analysis: The statistical analysis is robust. Normality check, correlational analysis and structural equation models with adequate adjustment indexes are performed.

Response 3

Thank you very much for your comment

Comment 4

Results: Results are presented in tables with correlation values, normality and standardized weights. Good interpretation of the effects according to the intensity of physical activity.

Response 4

Thank you very much for your comment

Comment 5

Discussion: The findings are adequately related to previous studies, although there are speculative interpretations without direct measurement. It is recommended: to explicitly state that these are untested hypotheses and to reduce excessively narrative or theoretical paragraphs without direct empirical basis in the study.

Response 5

Thank you very much for your comment. The discussion has been rephrased.

Comment 6

Limitations, conclusions and bibliographical references: Acknowledges limitations such as the type of sample and the online format of the questionnaire. It would be positive to highlight more the impossibility of establishing causality and to propose longitudinal or experimental studies as an alternative. The conclusions are in line with the objectives and results. However, the statement on medication consumption by physically active people should be more cautious or nuanced, as there is no measurement of the time of exercise or associated clinical variables. Relevant and updated references, recent and high impact citations.

Round 2

Reviewer 1 Report

Comments and Suggestions for Authors

Thank you for addressing all comments in detail.

Author Response

Thank you.

Reviewer 2 Report

Comments and Suggestions for Authors

Dear authors,

Thank you very much for taking my comments into account and for adding relevant information to your manuscript. Your manifest effort to clarify and deepen your methodological approach, in particular by adding details relating to the context of the sample (discipline, period of data collection) and the detailed characterization of the physical activities studied, is very much appreciated. These details significantly strengthen the scope of your results and the scientific rigor of your study.

I also welcome the fact that you have taken on board the recommendations aimed at improving the readability and accessibility of the text for non-specialist readers, in particular by clarifying the technical indicators and providing additional descriptive information.

In view of the changes made and subject to the favorable opinion of any other reviewers, I see no obstacle to the publication of your article.

Yours faithfully

Author Response

Thank you.